# Increase in Vascular Function Parameters According to Lifestyles in a Spanish Population without Previous Cardiovascular Disease—EVA Follow-Up Study

**DOI:** 10.3390/nu15214614

**Published:** 2023-10-30

**Authors:** Alicia Navarro Cáceres, Elena Navarro-Matías, Marta Gómez-Sánchez, Olaya Tamayo-Morales, Cristina Lugones-Sánchez, Susana González-Sánchez, Emiliano Rodríguez-Sánchez, Luis García-Ortiz, Leticia Gómez-Sánchez, Manuel A. Gómez-Marcos

**Affiliations:** 1Primary Care Research Unit of Salamanca (APISAL), Salamanca Primary Care Management, Institute of Biomedical Research of Salamanca (IBSAL), 37005 Salamanca, Spain; alicia.nav@usal.es (A.N.C.); enavarro@saludcastillayleon.es (E.N.-M.); olayatm@usal.es (O.T.-M.); crislugsa@gmail.com (C.L.-S.); gongar04@gmail.com (S.G.-S.); emiliano@usal.es (E.R.-S.); lgarciao@usal.es (L.G.-O.);; 2Castilla and León Health Service-SACYL, Regional Health Management, 37005 Salamanca, Spain; 3Home Hospitalization Service, Marqués de Valdecilla University Hospital, 39008 Santander, Spain; martagmzsnchz@gmail.com; 4Research Network on Chronicity, Primary Care and Health Promotion (RICAPPS), 37005 Salamanca, Spain; 5Department of Medicine, University of Salamanca, 28046 Salamanca, Spain; 6Department of Biomedical and Diagnostic Sciences, University of Salamanca, 37007 Salamanca, Spain; 7Emergency Service, University Hospital of La Paz, 37007 Madrid, Spain

**Keywords:** lifestyle, mediterranean diet, arterial stiffness, longitudinal study

## Abstract

The aim of this longitudinal descriptive observational study was to analyze the influence of different lifestyles on arterial stiffness (AS) throughout five years of follow-up and to describe the differences by sex in a Spanish adult population without cardiovascular disease at the start of the study. A random stratified sampling by age and sex was used to obtain 501 subjects included in the initial assessment. No cardiovascular disease was allowed in the subjects. The average age was 55.9 years, and 50.3% were women. A total of 480 subjects were analyzed again five years later. Alcohol and tobacco consumption were collected with standardized questionnaires. Adherence to the Mediterranean diet was assessed with the Mediterranean diet adherence screener (MEDAS) questionnaire. Physical activity was assessed with the short version of the International Physical Activity Questionnaire-Short Form (IPAQ-SF) and sedentary time was assessed with the Marshall Sitting Questionnaire (MSQ). AS was assessed by measuring carotid–femoral pulse wave velocity (cfPWV) and central augmentation index (CAIx) with SphygmoCor System^®^, and ankle pulse wave velocity (baPWV) and cardio ankle vascular index (CAVI) with Vasera VS-1500^®^. Increases in vascular function measures per year of follow-up were: cfPWV = 0.228 ± 0.360 m/s, baPWV = 0.186 ± 0.308 m/s, CAVI = 0.041 ± 0.181 m/s, and CAIx = 0.387 ± 2.664 m/s. In multiple regression analysis, positive association was shown between an increase in baPWV and tobacco index (β = 0.007) and alcohol consumption (β = 0.005). Negative association was shown between CAVI and Mediterranean diet score (β = −0.051). In multinomial logistic regression analysis, the OR of tobacco index of subjects with a cfPWV increase >P75 was OR = 1.025 and of subjects classified between P25 and P75 was OR = 1.026 regarding subjects classified with an increase <P25. The OR for alcohol consumption of participants with a baPWV increase >P75 was OR = 1.006 regarding subjects classified with an increase <P25. An OR = 0.838 was found in the Mediterranean diet score of subjects with an increased CAVI > P75, and an OR = 0.841 was found of subjects classified between P25–75 regarding subjects classified with an increase <P25. In conclusion, the association of lifestyle between an increase in AS measures at 5 year follow-up differs according to the AS measure analyzed. A positive association was shown with increased cfPWV and tobacco index, as well as alcohol consumption with increased baPWV. However, a negative association with CAVI and adherence to the Mediterranean diet was found.

## 1. Introduction

Arterial stiffness (AS) reflects the loss of elasticity of the arteries, affecting their ability to expand and contract [1,2]. AS is a predictor of cardiovascular disease risk similar or greater than traditional cardiovascular risk factors (CRFs) [3]. Thus, a positive association with cardiovascular events was described between AS measured noninvasively with carotid femoral pulse wave velocity (cfPWV), ankle pulse wave velocity (baPWV), cardio-ankle vascular index (CAVI), or central augmentation index (CAIx) [4,5,6,7,8]. AS is mainly determined by age, sex, and blood pressure [1,3]. 

On the other hand, unhealthy lifestyles may contribute to the development and worsening of AS. The relationship between AS and alcohol consumption is complex and depends on several factors, including the amount, acute or chronic alcohol consumption pattern, and individual susceptibility. Thus, acute alcohol consumption has vasodilatory properties that may temporarily decrease vascular resistance. Likewise, consumption may decrease AS related to antioxidant components, such as the polyphenols in red wine. However, chronic heavy alcohol consumption has negative effects on AS and is associated with increased risk of hypertension, vascular damage, plaque formation in arteries, and development of cardiovascular disease. This association has been exposed in several studies and meta-analyses where alcohol consumption has been independently associated with J-shaped AS [9,10,11]. Tobacco use increases vascular AS by altering endothelial homeostasis. This leads to reduced nitric oxide, increased altered lipid metabolism, and increased insulin resistance, inflammation, and oxidative stress [12,13,14]. The influence of diet on AS is unclear and some work suggests that diet quality may not function as an independent risk factor for AS [15]. Others suggest that a Mediterranean diet may help maintain arterial elasticity and is effective in improving cardiovascular health with clinically relevant reductions in blood pressure and AS [16]. Finally, studies reveal that increased physical activity and decreased sedentary time improve blood vessel flexibility and elasticity, and decrease AS, suggesting that regular physical activity is an effective index of AS [17,18,19,20].

However, we should not forget that lifestyle factors interact with each other and with other risk factors, such as genetics and age, to determine the degree of AS. Therefore, adopting a healthy lifestyle, including a balanced diet, regular physical activity, avoiding smoking, and daily consumption alcohol up to 20 g per day for women and 40 g for men, can play a crucial role in maintaining arterial elasticity and preventing cardiovascular disease. To the best of our knowledge, no prospective studies have analyzed the influence of these four lifestyle factors on AS in an adult population without previous cardiovascular disease over a five-year follow-up.

Therefore, the aims of this study are: (a) to analyze the influence of different lifestyles on the increase in cfPWV, baPWV, CAVI, and CAIx values during a five-year follow-up and (b) to describe the differences by sex in the Spanish adult population without cardiovascular disease at the start of the study.

## 2. Materials and Methods

### 2.1. Design

Longitudinal descriptive study of subjects included in the association between different risk factors and vascular accelerated ageing study (EVA study) [21]. This study is registered at ClinicalTrials.gov. Identifier NCT02623894.

### 2.2. Study Population

Urban population assigned to 5 urban primary care centers in Salamanca. A total of 501 subjects were selected using random sampling with stratified replacement by age groups (35, 45, 55, 65, and 75 years) and sex; 100 in each of the groups (50 men, 50 women), aged between 35 and 75 years—reference population 43,946. Subjects were included in the study between June 2016 and November 2017 and followed up with between May 2021 and October 2022. Inclusion criteria were age 35–75 years and signed informed consent. Exclusion criteria were terminally ill subjects, inability to travel to primary care centres, history of cardiovascular disease, glomerular filtration rate less than 30%, chronic inflammatory disease or an acute inflammatory process in the last three months, or being treated with estrogen, testosterone, or growth hormone. 

To detect an increase between baseline and follow-up measurement of 0.35 m/s assuming a standard deviation of ±1.70 m/s would require 206 subjects in each group for cfPWV. To detect an increase between the initial and follow-up measurement of 0.30 m/s assuming a standard deviation of ±1.51 m/s would require 221 subjects in each group for baPWV. To detect an increase between baseline and follow-up measurement of 0.20 units assuming a standard deviation of ±0.88 units would require 169 subjects in each group for CAVI. To detect an increase between baseline and follow-up of 2.5 units assuming a standard deviation of ±12 units would require 201 subjects in each group for CAVI. Accepting a 10% loss to follow-up and an alpha risk of 0.05 and a beta risk of 0.2 in a bilateral contrast.

### 2.3. Variables and Measuring Instruments

The variables collected and tests performed have been previously published in the EVA study protocol [21]. The professionals who performed the tests and questionnaires followed a standardized protocol.

#### 2.3.1. Measurement of Arterial Stiffness

The cfPWW and CAIx were estimated using the SphygmoCor device (AtCor Medical Pty Ltd., head office, West Ryde, Australia) by analyzing the carotid and femoral artery pulse waves, with the patient in the supine position, estimating the time delay compared to the ECG r-wave and calculating the cfPWV. Distances were measured with a tape measure from the sternal notch to the point where the sensor was placed in the carotid and femoral arteries. With the patient seated and with the arm resting on a rigid surface, pulse wave analysis was performed with a sensor on the radial artery, using a mathematical transformation to estimate the aortic pulse wave and CAIx adjusted to the heart rate of 75 beats [22]. 

CAVI and baPWV were estimated using the VaSera VS-1500 device (Fukuda Denshi Co, Ltd., Tokyo, Japan) according to the manufacturer’s instructions. CAVI values were automatically calculated by substituting stiffness parameters into the following equation to detect vascular elasticity and baPWV: stiffness parameter β = 2ρ × 1/(Ps − Pd) × ln (Ps/Pd) × PWV2 where ρ is blood density, Ps and Pd are SBP and DBP in mmHg, and PWV was measured between the aortic valve and the ankle. Cuffs were placed on arms and legs and a heart sound microphone was attached with double-sided tape to the sternum at the second intercostal space. Participants remained still and silent for 5 min. Only CAVI measurements obtained for at least 3 consecutive heartbeats were considered valid [23]. The baPWV was estimated using the following equation: baPWV = ((0.5934 × height(cm) + 14.4724))/tba, where tba is the time interval between the arm and ankle waves [24]. 

#### 2.3.2. Lifestyles Assessment

Tobacco use was assessed with a standardized questionnaire, indicating whether the participant was a smoker or non-smoker, the number of cigarettes consumed, and the number of years of smoking. Participants were defined as smokers if they smoked at the time of assessment or had stopped smoking within the last year. The tobacco index was determined by multiplying the daily consumption of cigarettes consumed with the number of smoking years, and dividing the result by 20 [25].

Alcohol consumption was evaluated using a standardized questionnaire, recording the quantity and type of alcohol consumed during one week, measured in g/week. It was considered low risk if the amount was less than 70 g/week for women and less than 140 g/week for men; moderate consumption if the amount was between 70 and 140 g/week for women and between 140 and 210 g/week for men; and risky consumption was considered if these intakes were higher than 140 g/W for women and 210 g/week for men [26]. 

Adherence to the Mediterranean diet was assessed with the 14-question Mediterranean diet adherence screener (MEDAS) questionnaire used in the prevention with Mediterranean diet (PREDIMED) study, validated in Spain. The questionnaire consists of 12 questions on food consumption frequency and 2 questions on dietary habits, each question is evaluated as 0 or 1, and the total score ranges from 0 to 14 [27]. 

Sedentary time was assessed with the Marshall Sitting Questionnaire (MSQ) [28]. This questionnaire is validated and assesses sitting time in hours and minutes on weekdays and weekends in five domains: transport, work, TV watching, home computer use, and leisure not specified in other domains. The total daily sedentary time will be calculated by summing the sitting time for each domain [29]. 

Physical activity was assessed with the short version of the questionnaire International Physical Activity Questionnaire—Short Form (IPAQ-SF) [30]. It consists of nine items, classifying physical activity during the last 7 days into three levels of intensity: (1) intense physical activity, (2) moderate activity, and (3) light activity. It is a self-report questionnaire that assesses sitting and active time in the last 7 days, differentiating between walking, moderate intensity, and vigorous intensity activities according to the estimated energy expenditure for each of them [3.3, 4.0, and 8.0 metabolic equivalents (METs), respectively]. 

#### 2.3.3. Evaluation of Cardio-Vascular Risk Factors

Clinical blood pressure was measured with a validated OMRON model M10-IT sphygmomanometer (Omron Health Care, Kyoto, Japan). Measurements were carried out according to the recommendations of the European Society of Hypertension [2]. Mean arterial pressure (MAP) was calculated with the following equation: MAP = (2 × diastolic blood pressure + systolic blood pressure)/3. Body weight was measured twice with a certified electronic scale (Seca 770, Birmingham, UK), which was calibrated (accuracy ±0.1 kg). Height was measured with a measuring rod (Seca 222, Birmingham, UK), and the average of two measurements was recorded. Body mass index was calculated as weight (kg) divided by height in m^2^. Plasma blood glucose, total cholesterol, high-density lipoprotein cholesterol, and triglycerides were determined in a venous blood sample at the Salamanca Primary Care Research Unit, performed between 08:00 and 09:00 h, fasting and without having smoked or consumed alcohol or caffeinated beverages during the previous 12 h, using standard automated enzymatic methods. Low-density lipoprotein cholesterol was determined using Friedewald formula. All analytical tests were processed in the same laboratory. A person was considered to have hypertension if they were on antihypertensive treatment or had blood pressure ≥140/90 mmHg. Participants were considered diabetics if they were on treatment with hypoglycemic agents or had blood glucose levels ≥126 mg/dL or HbA1c ≥ 6.5%; considered to have dyslipidemia if they were on treatment with lipid-lowering drugs or had fasting total cholesterol ≥ 240 mg/dL or low-density lipoprotein cholesterol ≥ 160 mg/dL or high-density lipoprotein cholesterol ≤40 mg/dL in men and ≤50 mg/dL in women or triglycerides ≥200 mg/dL; subjects with a body mass index ≥30 were classified as obese [21]. 

### 2.4. Statistical Analysis

Mean values of continuous variables are shown as mean ± standard deviation, and categorical variables are presented as numbers and percentages. Comparison of means between two independent groups was performed using Student *t*-test. Comparison of means of more than two groups was performed with analysis of variance (ANOVA), and ANCOVA was used to compare two adjusted means. Comparison of categorical variables was performed with the χ^2^ test. The correlation between the increase stiffness measures and the different lifestyles was performed using Pearson’s correlation coefficient. To analyze the association of the increase in AS measures with the different lifestyles, we performed several multiple linear regression analysis models, using as dependent variables the increase in stiffness measures (cfPWV, baPWV, CAVI, and CAIx), and as independent variables alcohol consumption in g/week, tobacco index, Mediterranean diet score, hours sitting per week, and METS/min/week. Finally, as adjustment variables age in years, sex (women = 0 and men = 1), mean blood pressure in mmHg, and antihypertensive, hypoglycemic, and lipid-lowering drugs (no consumption = 0 and yes consumption = 1) were used. To explore the association between percentile increase in AS measures and lifestyle, we performed several multinomial logistic regression models, using as dependent variables the increase in stiffness measures (cfPWV, baPWV, CAVI, and CAIx), in 3 categories (coded as percentile <25 = 1, between percentile 25–75 = 2, and percentile >75 = 3), taking as reference value percentile <25. As independent variables and as adjustment variables we used the same as in the multiple regression. All analyses were performed with the full sample and by sex. In the hypothesis test, statistical significance was set at α = 0.05. All analyses were performed with SPSS software for Windows, v28.0 (IBM Corp., Armonk, NY, USA).

### 2.5. Ethical Principles

This Project was approved by the committee of ethics of research with medicines of the health area of Salamanca, the baseline assessment dated on 4 May 2015, and the follow-up assessment on 13 November 2020 (CEIm reference code. PI 2020 10 569). Before the start of the study all participants signed the informed consent form. The standards of the Declaration of Helsinki were followed during the conduct of the study [31] and the WHO standards for observational studies were followed. Subjects were informed of the aims of the project and the risks and benefits of the examinations performed. The study did not contemplate any intervention involving a risk greater than the minimal risk involved in performing the different tests. All information generated in this study was stored, coded, and used exclusively for the purposes specified here. Both the samples and the data collected are associated with a code, kept under appropriate security conditions and it is guaranteed that the subjects cannot be identified through means considered reasonable by persons other than those authorized. The confidentiality of the subjects included has been guaranteed at all times in accordance with the provisions of Organic Law 3/2018, of 5th December, on the Protection of Personal Data and guarantee of digital rights and Regulation (EU) 2016/679 of the European Parliament and of the Council of 27 April 2016 on Data Protection (RGPD).

## 3. Results

### 3.1. Study Population

The flow chart of the recruitment progress at the start of the study and the selection of at the five-year follow-up, included, excluded, causes of exclusion, and loss to follow-up categorized by age group and gender can be viewed in Appendix A. A total of 501 subjects (50.3% female) were included in the study and at five-year follow-up 480 subjects (50.6% female) were evaluated. The characteristics of subjects who either died or were lost to follow-up can be located in Appendix A. Ten subjects (six males and four females) died during the follow-up period and we were unable to contact eleven subjects (six males and five females). 

Clinical variables, lifestyles, CRF at baseline assessment, globally, and grouped by sex are located in Table 1. Men consumed more alcohol, were more physically active, spent more time sitting, and were less adherent to the Mediterranean diet than women. Throughout the 5 years of follow-up, the increase in cfPWV was greater in men than in women (*p* = 0.040), with no differences between sexes in the rest of the AS measures analyzed. Increases in vascular function measures per year of follow-up were: cfPWV = 0.228 ± 0.360 m/s, baPWV = 0.186 ± 0.308 m/s, CAVI = 0.041 ± 0.181, and CAIx = 0.387 ± 2.664. 

### 3.2. Correlation between Increased Stiffness Measures and Lifestyles

Pearson’s correlation coefficients between increases in AS measures and lifestyles globally and grouped by sex are shown in Table 2. In the global analysis, baPWV shows positive correlation with alcohol consumption and tobacco index (r = 0.223 and r = 0.145) and negative correlation with hours of sitting per week (r = −0.098). CAVI is negatively correlated with adherence to the Mediterranean diet (r = −0.096).

### 3.3. Increased Measures of Stiffness in Subjects with Unhealthy and Healthy Lifestyles Globally and by Gender

Table 3 illustrates the differences in increases in AS measures between subjects with unhealthy and healthy lifestyles globally and gathered by sex. In the global analysis, the increase in baPWV is higher in drinkers compared to non-drinkers (*p* = 0.026), as well as in those who sit for less than 40 h per week (*p* = 0.011). No differences are found in the analysis by sex.

The global mean increase in estimated AS measures adjusted for age and mean blood pressure in subjects with both healthy and unhealthy lifestyles is located in Figure 1. The results grouped by sex are shown in Appendix A (males) and Appendix A (females). In the global analysis, the increase in baPWV is higher in drinkers than non-drinkers (*p* ≤ 0.001). The analysis by sex shows the increase in baPWV is higher in drinkers than non-drinkers in both men (*p* = 0.006) and women (*p* = 0.024).

The global results and grouped by sex of mean values of the different lifestyles of participants with lower 25 percentile increments, between the 25–75th percentile, and above the 75 percentile of the four measures analyzed are shown in Table 4. Higher CAVI increments are associated with lower adherence to the Mediterranean diet (*p* = 0.012). No difference is found in the analysis by sex. 

### 3.4. Association between Increased Measures of Stiffness and Lifestyles

Multiple regression analysis is located in Table 5. The increase in baPWV shows positive association with tobacco index β = 0.009 CI 95% (0.001–0.018) and with alcohol consumption β = 0.005 CI 95% (0.003–0.007). Additionally, a negative association is found between CAVI and the Mediterranean diet score β = −0.051 CI 95% (−0.092 to −0.010). The analysis by sex in males (Appendix A) maintains the positive association of baPWV with tobacco index (*p* = 0.032) and with alcohol consumption (*p* = 0.003). In women (Appendix A), it maintains the positive association of baPWV with alcohol consumption (*p* = 0.001) and the negative association with the Mediterranean diet score (*p* = 0.017).

The results of the multinomial logistic regression analysis are presented in Figure 2. The OR for tobacco index of individuals with an elevated cfPWV > P75 is OR = 1.025 (95% CI: 1.002–1.049), and for those classified between P25–P75 it is OR = 1.026 (95% CI: 1.002 – 1.050) in comparison to subjects classified with an increase of <P25. The OR for alcohol consumption in subjects with a baPWV increase of >P75 is OR = 1.006 (95% CI 1.002 – 1.010) when compared to subjects classified with an increase of <P25. The OR of the Mediterranean diet score in subjects with a CAVI of increase >P75 is OR = 0.838 (95% CI: 0.740 – 0.948), and for subjects classified between P25–P75 is OR = 0.841 (95% CI: 0.745 – 0.950) in respect to participants classified with an increase of < P25. Results stratified by sex are displayed in Appendix A (males) and Appendix A (females).

## 4. Discussion

To the best of our knowledge, this is the first study to analyze the impact of lifestyle on the increase in stiffness measures over 5 years. It found a positive association of tobacco index with increased cfPWV and tobacco index and alcohol consumption with baPWV, and a negative association of the Mediterranean diet compliance score with CAVI.

From a clinical perspective, the study of AS is a crucial topic, as it is a predictor of morbidity and mortality, regardless of traditional risk factors such as hypertension, diabetes, dyslipidemia, and smoking [7,8]. Therefore, identifying the determinants of AS could lead to better management and more efficient prevention of CV diseases.

The results of this work suggest a positive association between increased AS and alcohol consumption with baPWV in multiple regression analysis. However, this association does not hold in multinomial logistic regression analysis. Previously, published studies found discrepant results. Thus, several cross-sectional studies suggest a J-shaped relationship between alcohol consumption and AS to measure baPWV [32,33,34] and cfPWV [11,35]. However, the studies by Kim et al. [36], de Sluyter et al. [37], and Hwang et al. [38] found a positive linear association between different AS parameters and alcohol consumption. In the review published in 2022 [10], which analyses the relationship between AS and alcohol consumption, it is concluded that higher alcohol consumption is associated with worse AS values using different measurement methods (oscillometry or tonometry). It produced similar outcomes among the different populations studied (European, American, and Asian) and for almost all kinds of alcoholic beverages consumed. Thus, the debate on the correlation between alcohol consumption and AS remains open, as the relationship is complex and potentially affected by several factors such as type of alcohol, consumption levels, and gender and age differences. In contrast to other work suggesting that the association between alcohol consumption and AS is stronger in men [10], we found no gender differences in this study. The discrepancies can be explained by the fact that the studies analyzed are heterogeneous, using different methods of collection, measurement of AS, and analysis. In addition, most of them are cross-sectional studies. The beneficial effects of daily consumption alcohol up to 20 g per day for women and 40 g for men on AS may be explained due to the increase in HDL-cholesterol [39] and decreased insulin resistance [40]. On the other hand, high alcohol consumption leads to increased blood pressure and increased AS [41]. Finally, we cannot forget that alcohol consumption is one of the main risk factors for increased morbidity and mortality in the world, increasing the risk of all-cause mortality and cancer mortality as consumption increases [42]. Therefore, given the importance of AS as a predictor of cardiovascular risk, reducing alcohol intake in heavy drinkers should be recognized as a key dietary factor in improving cardiovascular health and disease prevention.

The results of this work suggest a positive association between increased AS and tobacco index with baPWV in multiple regression analysis and with cfPWV in multinomial logistic regression analysis. Supporting these results, several studies have shown increased AS in smokers [43,44]. This negative effect of smoking on AS has been found in acute, chronic, and passive smoking [45]. However, there are some studies that reveal no significant changes in the chronic effect of smoking on AS [10]. Results from the Gutenberg Health Study published in 2023 conclude that chronic smoking is strongly and dose-dependently associated with increased SA in a large population-based cohort, irrespective of sex, but with a stronger association in men [46]. However, although they analyzed data from 15,000 patients, they used the stiffness index and the augmentation index as measures of stiffness. On the contrary, other works suggest that the association between tobacco use and AS is generally more pronounced in men, indicating different patterns of vascular system stiffness in men and women [13]; we find no differences between sexes. This increase in AS with smoking is due to increased blood pressure and heart rate but also to the effect of nicotine, decreasing arterial wall elasticity, increasing inflammation and oxidative stress, and altering endothelial function [47]. 

The results of this study suggest a negative association between the increase in AS assessed by CAVI and the Mediterranean diet compliance score, that is, a higher consumption of a Mediterranean diet would be associated with a decrease in CAVI. These results align with those presented in a clinical trial (new dietary strategies addressing the specific needs of elderly population for healthy aging in Europe (NU-AGE)) [16], which concludes that a Mediterranean-style diet is effective in enhancing cardiovascular health with clinically relevant reductions in blood pressure and AS. They also observed an improvement in CAIx with a decrease of −12.4, without change in cfPWV in an Australian population [48]. These differences may be explained by the fact that elevated blood pressure accelerates stiffness in the conducting artery, as assessed by CAIx, but not the aortic stiffness, as evaluated by cfPWV [49]. 

In this study, we found no relationship between the increase in AS with physical activity and sedentary time. These results disagree with previous publications that have revealed that habitual physical activity and physical training decrease the increase in AS in healthy subjects or those with other CRF, whether assessed subjectively by questionnaires or objectively with an accelerometer, suggesting that AS may have a negative association with habitual physical activity [17,18,19,20,50,51]. Similarly, a meta-analysis reviewing 38 clinical trials with a total of 2089 patients concluded that aerobic and resistance exercise decreased AS [52]. Along these lines, several studies have observed a negative relationship between sedentary time and measures of stiffness [53,54,55]. Ahmadi et al. [56] showed that limiting sedentary time was associated with slower progression of aortic stiffness. Discrepancies with the results of this work may be due to the assessment of stiffness using questionnaires. 

### Limitations and Strengths

The present study has several limitations: 1. the findings are limited to adults without cardiovascular disease at baseline; 2. the sample studied is from an urban population. Therefore, the results may not be extrapolated to a rural population; 3. lifestyles were collected through questionnaires and could be susceptible to information bias. 

Nevertheless, this study also has several strengths: 1. it is the first study to analyze the four lifestyles jointly in a population sample; 2. it has analyzed AS with four different measures, which allows us to evaluate both central and peripheral AS; 3. it is a longitudinal study with a 5-year follow-up.

## 5. Conclusions

The influence of lifestyle on the increase in stiffness measures over 5 years differs according to the measure analyzed. The association between tobacco index and increase in cfPWV and both tobacco index and alcohol consumption with baPWV was positive. However, the Mediterranean diet was negatively associated with CAVI and CAIx.

## Figures and Tables

**Figure 1 nutrients-15-04614-f001:**
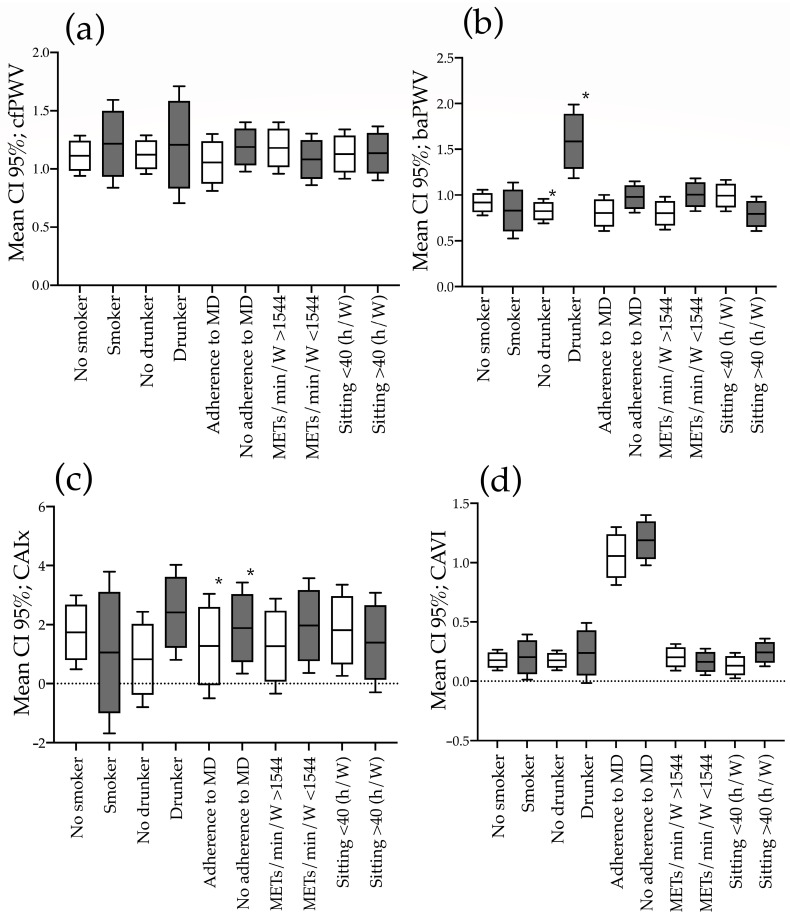
Increase in estimated arterial stiffness measures according to healthy or unhealthy lifestyles in men. ANOVA one way analysis. (**a**) cfPWV; (**b**) baPWV; (**c**) CAIx; (**d**) CAVI. CI: confidence interval; MD: Mediterranean diet; METs/min/W: basal metabolic rate/minutes/week; h/W: hours/week; cfPWV: carotid femoral pulse wave velocity; baPWV: brachial ankle pulse wave velocity; CAVI: cardiac ankle vascular index; CAIx: central augmentation index. *: *p* < 0.05.

**Figure 2 nutrients-15-04614-f002:**
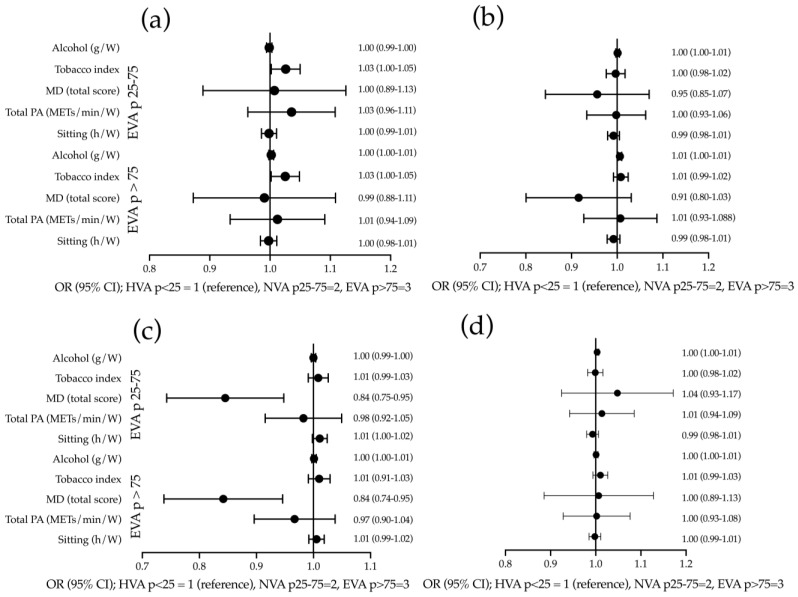
Association of lifestyles with increase in stiffness measures values in global. Multinomial logistic regression analysis: (**a**) using cfPWV; (**b**) using baPWV; (**c**) using CAIx; and (**d**) using CAVI as dependent variables. Tobacco consumption, alcohol consumption, Mediterranean diet score, total physical activity, and hours of sitting per week are used as independent variables and age, sex, mean blood pressure, and consumption of antihypertensive drugs are used as adjustment variables. cfPWV: carotid–femoral pulse wave velocity; baPWV: brachial–ankle pulse wave velocity; CAVI: cardiac–ankle vascular index; CAIx: central augmentation index; g/W: grams per week; MD: Mediterranean diet; PA: physical activity; METs/min/W: basal metabolic rate/minutes/week; h/W: hours/week; OR: odd ratio; HVA: healthy vascular ageing; NVA: normal vascular ageing; EVA: early vascular ageing.

**Table 1 nutrients-15-04614-t001:** General characteristics of the subjects included globally and by sex.

	Global (*n* = 480)	Men (*n* = 237)	Women (*n* = 243)	*p* Value
Lifestyles				
Alcohol (g/W)	91.17 ± 85.79	113.21 ± 93.37	55.71 ± 65.43	0.010
Appropriate consumption, *n* (%)	431 (89.8)	204 (42.5)	227 (47.3)	0,010
Daily cigarettes (*n*/day)	14.42 ± 10.65	14.79 ± 11.58	14.09 ± 10.20	0.646
Tobacco index	21.19 ± 22.08	21.61 ± 17.43	20.79 ± 25.70	0.797
No Smoker, *n* (%)	396 (82.5)	192 (40.0)	204 (42.5)	0.403
MD (total score)	7.17 ± 2.08	6.71 ± 1.96	7.61 ± 2.09	<0.001
Adherence to MD, *n* (%)	208 (43.3)	86 (17.9)	122 (25.4)	0.002
Total PA (METs/m/W)	2536 ± 3307	3305 ± 3789	1786 ± 2549	<0.001
>1545 METs/m/W, *n* (%)	240 (50)	154 (32.1)	86 (17.9)	<0.001
Sitting (h/W)	42.18 ± 17.80	47.81 ± 16.68	36.68 ± 17.178	<0.001
<40 h/W sitting, *n* (%)	261 (54.4)	98 (20.4)	163 (34)	<0.001
Conventional risk factors				
Age, (years)	56 ± 14.20	55.97 ± 14.30	56.02 ± 14.127	0.968
SBP (mmHg)	120.80 ± 23.38	126.50 ± 19.87	115.24 ± 25.19	<0.001
DBP (mmHg)	75.63 ± 9.94	77.55 ± 9.11	73.76 ± 10.37	<0.001
MBP (mmHg)	90.69 ± 12.62	93.87 ± 11.17	87.59 ± 13.19	<0.001
Hypertension, *n* (%)	142 (29.6)	79 (16.5)	63(13.1)	0.089
Antihypertensive drugs, *n* (%)	92 (19.2)	47 (9.8)	45 (9.4)	0.729
Total cholesterol (mg/dL)	195.10 ± 32.84	192.78 ± 32.79	197.35 ± 32.80	0.127
LDL cholesterol (mg/dL)	115.53 ± 29.48	117.38 ± 30.19	113.75 ± 28.72	0.180
HDL cholesterol (mg/dL)	58.94 ± 16.26	53.19 ± 14.17	64.52 ± 16.24	<0.001
Triglycerides (mg/dL)	103.18 ± 53.84	112.49 ± 55.15	94.11 ± 51.03	<0.001
Dyslipidemia, *n* (%)	185 (38.5)	91 (19.09)	94 (19.6)	1.000
Lipid–lowering drugs, *n* (%)	100 (20.8)	49.4 (10.0)	52 (10.8)	0.822
FPG (mg/dL)	88.03 ± 16.76	90.12 ± 18.62	85.99 ± 14.46	0.007
HbA1c (%)	5.49 ± 0.55	5.54 ± 0.61	5.43 ± 0.47	0.041
Diabetes mellitus, *n* (%)	36 (7.5)	25 (5.2)	11 (2.3)	0.015
Hypoglycemic drugs, *n* (%)	33 (6.9)	22 (4.6)	11 (2.3)	0.047
Height, cm	165,15 ± 9.77	171.80 ± 7.46	158.67 ± 7.024	<0.001
Weight, kg	62.71 ± 13.78	79.72 ± 11.78	65.68 ± 11.96	<0.001
BMI (kg/m^2^)	26.57 ± 4.25	27.01 ± 3.54	26.16 ± 4.82	0.029
Obesity, *n* (%)	91 (19.0)	41 (8.5)	50 (10.4)	0.415
CVR SCORE scale (%)	4.21 ± 6.14	5.46 ± 7.20	2.98 ± 4.59	<0.001
SCORE > 5, *n* (%)	134 (27.9)	85 (17.7)	49 (10.2)	<0.001
Increasing in arterial stiffness values				
cfPWV (m/seg)	1.11 ± 1.70	1.27 ± 1.70	0.95 ± 1.68	0.040
baPWV (m/seg)	0.90 ± 1.51	0.79 ± 1.53	1.01 ± 1.50	0.104
CAVI	0.19 ± 0.88	0.23 ± 0.86	0.14 ± 0.90	0.248
CAIx	1.94 ± 12.71	2.25 ± 13.97	1.63 ± 11.37	0.592

Values are means ± standard deviations for continuous data and number and proportions for categorical data. Smoker: actual tobacco consumer or former smoker of less than one year. Risky alcohol consumption in women was ≥140 g/week and in men ≥210 g/week. Adherence to MD. Physical activity and hours sitting per week cut-off value of the median. SD: standard deviation; g/W: grams/week; PA: physical activity; METs/m/W: basal metabolic rate/minute/week; MD: Mediterranean diet; SBP: systolic blood pressure; DBP: diastolic blood pressure; MBP: mean blood pressure; LDL: low-density lipoprotein; HDL: high-density lipoprotein; FPG: fasting plasma glucose; HbA1c: glycosylated hemoglobin; BMI: body mass index; CVR: cardiovascular risk; cfPWV: carotid–femoral pulse wave velocity; baPWV: brachial–ankle pulse wave velocity; CAVI: cardiac–ankle vascular index; CAIx: central augmentation index. *p* value: differences between men and women.

**Table 2 nutrients-15-04614-t002:** Correlation between lifestyles and increasing of arterial stiffness and lifestyles.

Global	cfPWV	baPWV	CAVI	CAIx
Alcohol consumption (g/W)	0.017	0.223 **	0.011	−0.046
Tobacco index	0.086	0.145 *	0.061	0.037
MD (total score)	0.003	0.029	−0.096 *	0.026
Total PA (METs/min/W)	0.019	−0.075	−0.055	−0.001
Sitting (h/W)	−0.025	−0.098 *	0.062	0.000
Men				
Alcohol consumption (g/W))	−0.001	0.240 **	−0.011	−0.015
Tobacco index	0.032	0.244 *	0.069	0.016
MD (total score)	0.015	0.078	−0.116	−0.021
Total PA (METs/min/W)	0.042	−0.037	−0.044	−0.016
Sitting (h/W)	−0.053	−0.107	0.094	−0.018
Women				
Alcohol consumption (g/W)	0.018	0.256 *	0.017	−0.126
Tobacco index	0.128	0.090	0.060	0.060
MD (total score)	0.033	−0.048	−0.061	−0.021
Total PA (METs/min/W)	−0.069	−0.096	−0.108	0.010
Sitting (h/W)	−0.061	−0.051	0.007	0.003

cfPWV: Carotid–femoral pulse wave velocity; baPWV: brachial–ankle pulse wave velocity; CAVI: cardiac–ankle vascular index; CAIx: central augmentation index; g/W: grams per week; MD: Mediterranean diet; PA: physical activity; METs/m/W: basal metabolic rate/min/week; h/W: hours/week. Pearson Coefficient. * *p* < 0.05; ** *p* < 0.01.

**Table 3 nutrients-15-04614-t003:** Difference of increasing arterial stiffness between healthy and unhealthy lifestyles. Global and stratified by sex.

Variables	Global (*n* = 480)	Men (*n* = 237)	Women (*n* = 243)
cfPWV			
Smoker/non-smoker	0.051 (−0.349 to 0.452)	0.057 (−0.498 to 0.613)	0.011 (−0.569 to 0.591)
Drinkers/non-drinkers	−0.016(−0.519 to 0.487)	−0.195 (−0.823 to 0.433)	0.135 (−0.724 to 0.994)
No adherence/adherence to MD	0.007 (−0.299 to 0.315)	−0.099 (−0.552 to 0.353)	0.017 (−0.409 to 0.444)
Sedentary/actives	−0.082 (−0.387 to 0.222)	−0.046 (−0.502 to 0.410)	0.070 (−0.375 to 0.516)
Sitting > 40/sitting < 40 h/W	−0.060 (−0.366 to 0.245)	−0.181 (−0.638 to 0.275)	−0.123 (−0.576 to 0.330)
baPWV			
Smoker/non-smoker	−0.231 (−0.589 to 0.126)	−0.222 (−0.722 to 0.277)	−0.218 (−0.735 to 0.296)
Drinkers/non-drinkers	0.581 (0.134 to 1.028)	0.713 (0.154 to 1.272)	0.487 (−0.276 to 1.251)
No adherence/adherence to MD	−0.159 (−0.434 to 0.115)	−0.338 (−0.743 to 0.067)	0.058 (−0.322 to 0.4389)
Sedentary/actives	0.269 (−0.003 to 0.540)	0.042 (−0369 to 0.453)	0.395 (0.001 to 0.790)
Sitting > 40/sitting < 40 h/W	−0.317 (−0.589 to −0.045)	−0.349 (−0.744 to 0.047)	−0.200 (−0.604 to 0.204)
CAVI			
Smoker/non-smoker	0.067 (−0.140 to 0.274)	0.079 (−0.200 to 0.360)	0.043 (−0.266 to 0.352)
Drinkers/non-drinkers	0.041 (−0.220 to 0.301)	0.021 (−0.297 to 0.339)	0.025(−0.433 to 0.484)
No adherence/adherence to MD	0.102 (−0.057 to 0.261)	0.117 (−0.111 to 0.345)	0.067 (−0.160 to 0.294)
Sedentary/actives	−0.029 (−0.187 to 0.129)	−0.152 (−0.400 to 0.096)	0.144 (−0.093 to 0.381)
Sitting > 40/sitting < 40 h/W	0.083 (−0.075 to 0.241)	0.148 (−0.074 to 0.371)	−0.027 (−0.269 to 0.215)
CAIx			
Smoker/non-smoker	0.736 (−2.267 to 3.739)	2.549 (−2.007 to 7.105)	−1.348 (−5.267 to 2.572)
Drinkers/non-drinkers	0.755 (−1.526 to 3.037)	−0.149 (−3.848 to 3.551)	1.409 (−1.554 to 4.373)
No adherence/adherence to MD	1.359 (−0.941 to 3.660)	1.029 (−2.694 to 4.753)	1.538 (−1.336 to 4.411)
Sedentary/actives	0.665 (−1.617 to 2.947)	2.307 (−1.437 to 6.051)	−0.406 (−3.417 to 2.605)
Sitting > 40/sitting < 40 h/W	−0.307 (−2.598 to 1.984)	−0.365 (−4.003 to 3.272)	−0.649 (−3.712 to 2.414)

Values are means ± standard deviations for continuous data and number and proportions for categorical data. Smoker: actual tobacco consumer or former smoker of less than one year. Risky alcohol consumption in women was ≥140 g/week and in men ≥210 g/week. Adherence to MD: Score Value ≥ 9. Physical activity and hours spent sitting per week cut-off point for median value. cfPWV: carotid–femoral pulse wave velocity; baPWV: brachial–ankle pulse wave velocity; CAVI: cardiac–ankle vascular index; CAIx: central augmentation index; MD: Mediterranean diet; h/W: hours/week.

**Table 4 nutrients-15-04614-t004:** Analysis of lifestyles according to increasing arterial stiffness percentiles.

Global	<P25	Entre P25–75	>P75	*p*
cfPWV				
Alcohol (g/W)	90.64 ± 71.58	86.72 ± 85.97	101.67 ± 97.73	0.551
Tobacco index	16.93 ± 16.28	21.53 ± 25.02	25.01 ± 19.66	0.210
Mediterranean diet (total score)	7.19 ± 2.04	7.03 ± 2.07	7.41 ± 2.12	0.266
Total PA (METs/m/W)	2517 ± 3700	2476 ± 2886	2677 ± 3697	0.861
Sitting (h/W)	43.36 ± 17.29	41.76 ± 18.30	41.85 ± 17.39	0.709
baPWV				
Alcohol (g/W)	92.84 ± 73.51	77.48 ± 73.50	106.57 ± 103.10	0.066
Tobacco index	17.35 ± 16.82	22.32 ± 26.51	22.27 ± 18.06	0.426
Mediterranean diet (total score)	7.20 ± 2.07	7.05 ± 2.13	7.28 ± 2.02	0.558
Total PA (METs/m/W)	2605 ± 3773	2429 ± 2846	2615 ± 3491	0.838
Sitting (h/W)	43.69 ± 17.13	42.30 ± 18.77	41.03 ± 17.11	0.468
CAVI				
Alcohol (g/W)	92.11 ± 88.11	88.35 ± 87.85	95.16 ± 80.79	0.876
Tobacco index	18.85 ± 13.73	20.65 ± 26.75	25.37 ± 20.44	0.312
Mediterranean diet (total score)	7.65 ± 2.06	7.04 ± 2.11	6.94 ± 1.95	0.012
Total PA (METs/m/W)	2785 ± 4166	2599 ± 3206	21868 ± 2426	0.322
Sitting (h/W)	39.58 ± 19.41	42.39 ± 17.02	44.31 ± 17.50	0.144
CAIx				
Alcohol (g/W)	97.42 ± 93.91	96.30 ± 87.46	74.92 ± 71.81	0.231
Tobacco index	20.99 ± 13.33	20.10 ± 18.78	23.35 ± 33.51	0.711
Mediterranean diet (total score)	7.34 ± 1.86	7.21 ± 2.20	6.89 ± 1.97	0.242
Total PA (METs/m/W)	2196 ± 2363	2704 ± 3793	2534 ± 3012	0.381
Sitting (h/W)	42.72 ± 19.36	41.51 ± 17.02	43.06 ± 17,86	0.694
Men				
cfPWV				
Alcohol (g/W)	117.58 ± 75.40	103.35 ± 92.77	132.43 ± 107.41	0.294
Tobacco index	19.38 ± 17.71	20.02 ± 14.64	26.76 ± 21.93	0.245
Mediterranean diet (total score)	6.73 ± 2.02	6.60 ± 1.84	6.91 ± 2.16	0.603
Total PA (METs/m/W)	3048 ± 3720	3103 ± 3282	3883 ± 4635	0.353
Sitting (h/W)	47.86 ± 15.49	48.63 ± 16.75	46.23 ± 17.56	0.649
baPWV				
Alcohol (g/W)	121.21 ± 76.40	93.36 ± 80.99	130.09 ± 109.39	0.087
Tobacco index	19.38 ± 17.71	20.92 ± 15.13	23.46 ± 19.66	0.680
Mediterranean diet (total score)	6.76 ± 2.04	6.54 ± 1.86	6.86 ± 2.03	0.532
Total PA (METs/m/W)	3099 ± 3787	3174 ± 3311	3544 ± 4240	0.731
Sitting (h/W)	48.37 ± 16.60	49.57 ± 17.10	45.75 ± 16.75	0.282
CAVI				
Alcohol (g/W)	118.16 ± 100.04	113.86 ± 95.80	107.95 ± 85.24	0.884
Tobacco index	17.72 ± 14.69	23.90 ± 19.46	22.86 ± 16.94	0.311
Mediterranean diet (total score)	7.24 ± 1.86	6.70 ± 2.03	6.32 ± 1.84	0.035
Total PA (METs/m/W)	3271 ± 4501	3658 ± 3857	2749 ± 2967	0.291
Sitting (h/W)	45.66 ± 18.15	46.94 ± 16.40	50.93 ± 15.72	0.164
CAIx				
Alcohol (g/W)	124.31 ± 109.00	114.54 ± 93.62	99.31 ± 74.97	0.519
Tobacco index	21.65 ± 13.18	22.34 ± 20.61	20.32 ± 16.63	0.905
Mediterranean diet (total score)	6.75 ± 1.68	6.87 ± 2.06	6.34 ± 1.98	0.241
Total PA (METs/m/W)	3116 ± 2930	3265 ± 4163	3572 ± 3717	0.805
Sitting (h/W)	48.35 ± 19.15	47.06 ± 15.89	48.89 ± 16.03	0.762
Women				
cfPWV				
Alcohol (g/W)	55.83 ± 48.67	57.07 ± 62.92	52.73 ± 52.02	0.958
Tobacco index	15.32 ± 15.39	22.99 ± 32.08	22.90 ± 16.86	0.404
Mediterranean diet (total score)	7.55 ± 1.98	7.46 ± 2.20	8.00 ± 1.93	0.278
Total PA (METs/m/W)	2112 ± 3661	1848 ± 2273	1251 ± 916	0.166
Sitting (h/W)	39.93 ± 17.91	34.89 ± 17.23	36.67 ± 15.83	0.157
baPWV				
Alcohol consumption (g/W)	55.45 ± 50.06	54.40 ± 52.74	57.86 ± 67.46	0.969
Tobacco index	15.79 ± 16.30	23.36 ± 32.63	20.57 ± 15.70	0.488
Mediterranean diet (total score)	7.54 ± 2.03	7.51 ± 2.24	7.80 ± 1.91	0.621
Total PA (METs/m/W)	2221 ± 3747	1755 ± 2151	1786 ± 2549	0.219
Sitting (h/W)	40.06 ± 17.49	35.72 ± 17.83	36.68 ± 17.18	0.190
CAVI				
Alcohol (g/W)	54.04 ± 47.03	51.74 ± 58.79	67.00 ± 63.09	0.596
Tobacco index	20.07 ± 12.79	18.10 ± 31.27	27.99 ± 23.68	0.318
Mediterranean diet (total score)	7.98 ± 2.17	7.35 ± 2.14	7.75 ± 1.79	0.113
Total PA (METs/m/W)	2387 ± 3860	1625 ± 2036	1412 ± 1075	0.070
Sitting (h/W)	34.61 ± 19.11	38.21 ± 16.56	35.69 ± 15.99	0.347
CAIx				
Alcohol (g/W)	62.86 ± 54.56	60.77 ± 60.73	39.80 ± 50.20	0.255
Tobacco index	20.38 ± 13.66	18.10 ± 16.97	26.27 ± 44.31	0.443
Mediterranean diet (total score)	7.82 ± 1.88	7.54 ± 2.30	7.50 ± 1.80	0.619
Total PA (METs/m/W)	1770 ± 1388	2143 ± 3305	1377 ± 1166	0.081
Sitting (h/W)	38.10 ± 18.41	35.96 ± 16.35	36.57 ± 17.68	0.713

cfPWV: carotid–femoral pulse wave velocity; baPWV: brachial–ankle pulse wave velocity; CAVI: cardiac–ankle vascular index; CAIx: central augmentation index. g/W: grams per week; PA: physical activity; METs/m/W: basal metabolic rate/minute/week; h/W: hours/week. *p*: Differences between groups.

**Table 5 nutrients-15-04614-t005:** Association of increasing arterial stiffness with lifestyles in global. Multiple regression analysis.

cfPWV (m/s)	β	IC 95%	*p*
Tobacco index	0.007	(−0.003 to 0.016)	0.183
Alcohol consumption (gr/W)	0.001	(−0.002 to 0.003)	0.656
Mediterranean diet (total score)	−0.025	(−0.107 to 0.058)	0.555
Total PA (METs/min/W)	0.003	(−0.046 to 0.052)	0.899
Sitting (h/W)	−0.003	(−0.012 to 0.006)	0.523
baPWV (m/s)			
Tobacco index	0.009	(0.001 to 0.018)	0.047
Alcohol consumption (g/W)	0.005	(0.003 to 0.007)	<0.001
Mediterranean diet (total score)	−0.064	(−0.131 to 0.003)	0.061
Total PA (METs/min/W)	−0.010	(−0.050 to 0.030)	0.622
Sitting (h/W)	−0.005	(−0.013 to 0.002)	0.165
CAVI			
Tobacco index	0.003	(−0.004 to 0.009)	0.427
Alcohol consumption (g/W)	0.000	(−0.001 to 0.002)	0.711
Mediterranean diet (total score)	−0.051	(−0.092 to −0.010)	0.016
Total PA (METs/min/W)	−0.016	(−0.040 to 0.009)	0.215
Sitting (h/W)	0.003	(−0.002 to 0.007)	0.258
CAIx			
Tobacco index	0.078	(−0.009 to 0.165)	0.079
Alcohol consumption (g/W)	0.011	(−0.008 to 0.030)	0.263
Mediterranean diet (total score)	0.242	(−0.358 to 0.843)	0.428
Total PA (METs/min/W)	−0.024	(−0.381 to 0.333)	0.896
Sitting (h/W)	−0.015	(−0.082 to 0.052)	0.667

Multiple regression analysis using as dependent variables cfPWV, baPWV, CAVI, and CAIx. Independent variables are lifestyles (tobacco consumption, alcohol, Mediterranean diet score, total physical activity, and hours sitting per week) and adjustment variables are age, sex, mean arterial pressure, and consumption of antihypertensive drugs, hypoglycemic, and lipid-lowering agents. cfPWV: carotid–femoral pulse wave velocity; baPWV: brachial–ankle pulse wave velocity; CAVI: cardiac–ankle vascular index; CAIx: central augmentation index. m/s: meters/second; g/W: grams per week; PA: physical activity; METs/m/W: basal metabolic rate/minute/week; h/W: hours/week.

## Data Availability

The variables that we use in the analyses carried out to obtain the results of this work are available upon reasonable request to the corresponding author.

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
