# Peer review of "Increase in Vascular Function Parameters According to Lifestyles in a Spanish Population without Previous Cardiovascular Disease—EVA Follow-Up Study"

_nutrients, 2023, doi:10.3390/nu15214614_

Round 1

Reviewer 1 Report

Comments and Suggestions for Authors

The topic is interesting, but this version of the article seems to have been published previously, by M. Go´mez-Sa´nchez et al. / Rev Esp Cardiol. 2021;74(10):854-861.

If it is only my confusion, please argue that it is another original study.

Author Response

Reviewer 1.

The topic is interesting, but this version of the article seems to have been published previously, by M. Go´mez-Sa´nchez et al. / Rev Esp Cardiol. 2021;74(10):854-861.

If it is only my confusion, please argue that it is another original study.

Answer

The manuscript published in 2021 entitled Relationship of healthy vascular aging with lifestyle and metabolic syndrome in the general Spanish population. The EVA study” [1]

Our objective was to study the relationship of healthy vascular aging (HVA) with lifestyle and the components of metabolic syndrome. We also analyzed the differences between chronological age and heart age (HA) and vascular age (VA) in the Spanish adult population without cardiovascular disease.

The published results referred to cross-sectional study data collected in the initial analysis of the EVA study [2].

The aim of this longitudinal descriptive observational study was to analyse the influence of different lifestyles on arterial stiffness (AS) throughout five years of follow-up and to describe the differences by sex in a Spanish adult population without cardiovascular disease at the start of the study.

The results of this study analyze the association of lifestyles with increased measures of arterial stiffness during 5 years of follow-up (difference between the current evaluation and the initial evaluation).

References

  1. Gómez-Sánchez, M.; Gómez-Sánchez, L.; Patino-Alonso, M.C.; Alonso-Domínguez, R.; Sánchez-Aguadero, N.; Recio-Rodríguez, J.I.; González-Sánchez, J.; García-Ortiz, L.; Gómez-Marcos, M.A. Relationship of healthy vascular aging with lifestyle and metabolic syndrome in the general Spanish population. The EVA study. Rev Esp Cardiol (Engl Ed) 2021, 74, 854-861, doi:10.1016/j.rec.2020.06.040.
  2. Gomez-Marcos, M.A.; Martinez-Salgado, C.; Gonzalez-Sarmiento, R.; Hernandez-Rivas, J.M.; Sanchez-Fernandez, P.L.; Recio-Rodriguez, J.I.; Rodriguez-Sanchez, E.; García-Ortiz, L. Association between different risk factors and vascular accelerated ageing (EVA study): study protocol for a cross-sectional, descriptive observational study. BMJ Open 2016, 6, e011031, doi:10.1136/bmjopen-2016-011031.

Reviewer 2 Report

Comments and Suggestions for Authors

In the manuscript submitted to me for review entitled " Increase in vascular function parameters according to lifestyles in a Spanish population without previous cardiovascular disease. EVA follow-up studythe authors Alicia Navarro-Caceres, Elena Navarro-Matías, Marta Gomez-Sanchez, Olaya Tamayo-Morales, Cristina Lugones-Sanchez, Susana Gonzalez-Rodriguez, Emiliano Rodriguez-Sanchez, Luis García-Ortiz, Leticia Gomez-Sanchez and Manuel A Gomez-Marcos analyzed the influence of different lifestyles on arterial stiffness (AS) over a five-year follow-up period in a Spanish population without cardiovascular disease at baseline. The influence of alcohol consumption, smoking and adherence to the Mediterranean diet was evaluated.

The research conducted by the authors is extremely important for the control and prevention of arterial stiffness and the subsequent cardiovascular diseases it can lead to.

To support their research, the authors used 56 references that cover research for more than three decades, but the main information is from the last 5 years -29 references (more than ½ of the total number).

I have no objections to the authors regarding the very design of the research, the methods used and the statistical processing of the results. The research was conducted and described in an extremely consistent and detailed manner. I have only two suggestions for improving individual parts of the manuscript.

1. In my opinion, in figure 2, the inscriptions on the right side of each subfigure are of poor quality. They are blurry, and even if they are enlarged they are not readable. If it is possible to improve their quality to make them clearly visible.

2. Some of the references do not list all the authors (Nos. 2, 3, 4, 5, 6, 16, 18, 20, 23, 27, 46). Personally, when I read an article, I prefer the references to be fully written, instead of having to search for some of the authors. I think it would be helpful to your readers if all authors in all references are listed.

Author Response

Reviewer 2.

In the manuscript submitted to me for review entitled "Increase in vascular function parameters according to lifestyles in a Spanish population without previous cardiovascular disease. EVA follow-up study” the authors Alicia Navarro-Caceres, Elena Navarro-Matías, Marta Gomez-Sanchez, Olaya Tamayo-Morales, Cristina Lugones-Sanchez, Susana Gonzalez-Rodriguez, Emiliano Rodriguez-Sanchez, Luis García-Ortiz, Leticia Gomez-Sanchez and Manuel A Gomez-Marcos analyzed the influence of different lifestyles on arterial stiffness (AS) over a five-year follow-up period in a Spanish population without cardiovascular disease at baseline. The influence of alcohol consumption, smoking and adherence to the Mediterranean diet was evaluated.

The research conducted by the authors is extremely important for the control and prevention of arterial stiffness and the subsequent cardiovascular diseases it can lead to.

To support their research, the authors used 56 references that cover research for more than three decades, but the main information is from the last 5 years -29 references (more than ½ of the total number).

I have no objections to the authors regarding the very design of the research, the methods used and the statistical processing of the results. The research was conducted and described in an extremely consistent and detailed manner. I have only two suggestions for improving individual parts of the manuscript.

Answer

First of all, we would like to thank you for reviewing the manuscript.

  1. In my opinion, in figure 2, the inscriptions on the right side of each subfigure are of poor quality. They are blurry, and even if they are enlarged, they are not readable. If it is possible to improve their quality to make them clearly visible.

Answer

We have enlarged the inscriptions on the right side of each subfigure. As a result, the quality of these has increased. Also, if you consider them necessary, we can send you the figures in TIF format with higher quality.

  1. Some of the references do not list all the authors (Nos. 2, 3, 4, 5, 6, 16, 18, 20, 23, 27, 46). Personally, when I read an article, I prefer the references to be fully written, instead of having to search for some of the authors. I think it would be helpful to your readers if all authors in all references are listed.

Answer

We have followed the rules set by the journal Nutriens. We have used the bibliographic manager EDNOTE.

However, so that you can have the names of all the authors in the citations, we attach them to you in this answer. [1; 2; 3; 4; 5; 6; 7; 8; 9; 10; 11; 12]

References

[1] B. Williams, G. Mancia, W. Spiering, E. Agabiti Rosei, M. Azizi, M. Burnier, D.L. Clement, A. Coca, G. de Simone, A. Dominiczak, T. Kahan, F. Mahfoud, J. Redon, L. Ruilope, A. Zanchetti, M. Kerins, S.E. Kjeldsen, R. Kreutz, S. Laurent, G.Y.H. Lip, R. McManus, K. Narkiewicz, F. Ruschitzka, R.E. Schmieder, E. Shlyakhto, C. Tsioufis, V. Aboyans, and I. Desormais, 2018 ESC/ESH Guidelines for the management of arterial hypertension: The Task Force for the management of arterial hypertension of the European Society of Cardiology and the European Society of Hypertension: The Task Force for the management of arterial hypertension of the European Society of Cardiology and the European Society of Hypertension. J Hypertens 36 (2018) 1953-2041.

[2] K. Matsushita, N. Ding, E.D. Kim, M. Budoff, J.A. Chirinos, B. Fernhall, N.M. Hamburg, K. Kario, T. Miyoshi, H. Tanaka, and R. Townsend, Cardio-ankle vascular index and cardiovascular disease: Systematic review and meta-analysis of prospective and cross-sectional studies. J Clin Hypertens (Greenwich) 21 (2019) 16-24.

[3] T. Miyoshi, H. Ito, K. Shirai, S. Horinaka, J. Higaki, S. Yamamura, A. Saiki, M. Takahashi, M. Masaki, T. Okura, K. Kotani, T. Kubozono, R. Yoshioka, H. Kihara, K. Hasegawa, N. Satoh-Asahara, and H. Orimo, Predictive Value of the Cardio-Ankle Vascular Index for Cardiovascular Events in Patients at Cardiovascular Risk. J Am Heart Assoc 10 (2021) e020103.

[4] F.L.J. Visseren, F. Mach, Y.M. Smulders, D. Carballo, K.C. Koskinas, M. Bäck, A. Benetos, A. Biffi, J.M. Boavida, D. Capodanno, B. Cosyns, C. Crawford, C.H. Davos, I. Desormais, E. Di Angelantonio, O.H. Franco, S. Halvorsen, F.D.R. Hobbs, M. Hollander, E.A. Jankowska, M. Michal, S. Sacco, N. Sattar, L. Tokgozoglu, S. Tonstad, K.P. Tsioufis, I. van Dis, I.C. van Gelder, C. Wanner, and B. Williams, 2021 ESC Guidelines on cardiovascular disease prevention in clinical practice. Eur Heart J 42 (2021) 3227-3337.

[5] T. Ohkuma, T. Ninomiya, H. Tomiyama, K. Kario, S. Hoshide, Y. Kita, T. Inoguchi, Y. Maeda, K. Kohara, Y. Tabara, M. Nakamura, T. Ohkubo, H. Watada, M. Munakata, M. Ohishi, N. Ito, T. Shoji, C. Vlachopoulos, and A. Yamashina, Brachial-Ankle Pulse Wave Velocity and the Risk Prediction of Cardiovascular Disease: An Individual Participant Data Meta-Analysis. Hypertension 69 (2017) 1045-1052.

[6] A. Jennings, A.M. Berendsen, L. de Groot, E.J.M. Feskens, A. Brzozowska, E. Sicinska, B. Pietruszka, N. Meunier, E. Caumon, C. Malpuech-Brugère, A. Santoro, R. Ostan, C. Franceschi, R. Gillings, O.N. CM, S.J. Fairweather-Tait, A.M. Minihane, and A. Cassidy, Mediterranean-Style Diet Improves Systolic Blood Pressure and Arterial Stiffness in Older Adults. Hypertension 73 (2019) 578-586.

[7] E.J. Vandercappellen, R.M.A. Henry, H. Savelberg, J.D. van der Berg, K.D. Reesink, N.C. Schaper, S. Eussen, M. van Dongen, P.C. Dagnelie, M.T. Schram, M.M.J. van Greevenbroek, A. Wesselius, C.J.H. van der Kallen, S. Köhler, C.D.A. Stehouwer, and A. Koster, Association of the Amount and Pattern of Physical Activity With Arterial Stiffness: The Maastricht Study. J Am Heart Assoc 9 (2020) e017502.

[8] O. Hahad, V.H. Schmitt, N. Arnold, K. Keller, J.H. Prochaska, P.S. Wild, A. Schulz, K.J. Lackner, N. Pfeiffer, I. Schmidtmann, M. Michal, J.M. Schattenberg, O. Tüscher, A. Daiber, and T. Münzel, Chronic cigarette smoking is associated with increased arterial stiffness in men and women: evidence from a large population-based cohort. Clin Res Cardiol 112 (2023) 270-284.

[9] H. Schröder, M. Fitó, R. Estruch, M.A. Martínez-González, D. Corella, J. Salas-Salvadó, R. Lamuela-Raventós, E. Ros, I. Salaverría, M. Fiol, J. Lapetra, E. Vinyoles, E. Gómez-Gracia, C. Lahoz, L. Serra-Majem, X. Pintó, V. Ruiz-Gutierrez, and M.I. Covas, A short screener is valid for assessing Mediterranean diet adherence among older Spanish men and women. J Nutr 141 (2011) 1140-5.

[10] K. Shirai, N. Hiruta, M. Song, T. Kurosu, J. Suzuki, T. Tomaru, Y. Miyashita, A. Saiki, M. Takahashi, K. Suzuki, and M. Takata, Cardio-ankle vascular index (CAVI) as a novel indicator of arterial stiffness: theory, evidence and perspectives. J Atheroscler Thromb 18 (2011) 924-38.

[11] T. Ohkuma, T. Ninomiya, H. Tomiyama, K. Kario, S. Hoshide, Y. Kita, T. Inoguchi, Y. Maeda, K. Kohara, Y. Tabara, M. Nakamura, T. Ohkubo, H. Watada, M. Munakata, M. Ohishi, N. Ito, M. Nakamura, T. Shoji, C. Vlachopoulos, and A. Yamashina, Brachial-Ankle Pulse Wave Velocity and the Risk Prediction of Cardiovascular Disease: An Individual Participant Data Meta-Analysis. Hypertension 69 (2017) 1045-1052.

[12] S.J. Islam, N. Beydoun, A. Mehta, J.H. Kim, Y.A. Ko, Q. Jin, P. Baltrus, M.L. Topel, C. Liu, M.S. Mujahid, V. Vaccarino, M. Sims, K. Ejaz, C. Searles, S.B. Dunbar, T.T. Lewis, H.A. Taylor, P. Pemu, and A.A. Quyyumi, Association of physical activity with arterial stiffness among Black adults. Vasc Med 27 (2022) 13-20.

Reviewer 3 Report

Comments and Suggestions for Authors

Alicia Navarro Cáceres et al., the study focus on the increase in vascular function parameters according to lifestyles in a Spanish population without previous cardiovascular disease. EVA follow-up study. This work is written clearly. The results were analyzed according to appropriate methods. The discussion is carried out correctly. The conclusions presented are consistent.

Figures 1 and 2 are not visible letters.

Comments on the Quality of English Language

Minor editing of English language required

Author Response

Reviewer 3.

Alicia Navarro Cáceres et al., the study focus on the increase in vascular function parameters according to lifestyles in a Spanish population without previous cardiovascular disease. EVA follow-up study. This work is written clearly. The results were analyzed according to appropriate methods. The discussion is carried out correctly. The conclusions presented are consistent.

Figures 1 and 2 are not visible letters.

Answer

We have enlarged the inscriptions on the right side of each subfigure. As a result, the quality of these has increased. Also, if you consider them necessary, we can send you the figures in TIF format with higher quality.

Comments on the Quality of English Language. Minor editing of English language required

Answer

We have reviewed the manuscript. However, if you consider it necessary, we can edit it.

Round 2

Reviewer 1 Report

Comments and Suggestions for Authors

The arguments outlined highlight elements of differentiation and originality from the previously published works by the group of authors. However, I recommend the authors use a different formulation when they say "moderate alcohol consumption". I want a quantitative expression regarding alcohol consumption, rather than the qualitative expression preferred by the authors.

Author Response

Answer

Following their recommendations, we have added a quantitative measure on alcohol consumption.

Changes in the manuscript have been shaved in yellow

Line 80-86

However, we should not forget that lifestyle factors interact with each other and with other risk factors, such as genetics and age, to determine the degree of AS. Therefore, adopting a healthy lifestyle, including a balanced diet, regular physical activity, avoiding smoking and moderate alcohol consumption (daily consumption up to 20 g per day for women and 40 g for men), can play a crucial role in maintaining arterial elasticity and preventing cardiovascular disease.

Line  389-392

The beneficial effects of light, moderate alcohol consumption, (daily consumption up to 20 g per day for women and 40 g for men), on AS may be explained due to the increase in HDL-cholesterol [39] and decreased insulin resistance [40]. On the other hand, high alcohol consumption leads to increased blood pressure and increased AS  [41].